# OpenReview forum: "Geometric Conformal Outlier Synthesis"
_ICLR.cc/2026/Conference — Submitted to ICLR 2026_

### Official Review · Reviewer_WDSt · 2025-10-30

**Soundness:** 2
**Presentation:** 3
**Contribution:** 1
**Rating:** 2
**Confidence:** 3

**Summary:**

This paper proposes Geometric Conformal Outlier Synthesis (GCOS), a training-time regularization framework for improving out-of-distribution (OOD) robustness. GCOS combines geometric information from PCA with a conformal prediction heuristic to generate synthetic outliers in the latent space. The method is evaluated on several datasets and compared against the baseline Virtual Outlier Synthesis (VOS).

**Strengths:**

The paper considers the application of conformal prediction in the context of outlier synthesis, which is an interesting attempt to connect uncertainty quantification with OOD data generation.

**Weaknesses:**

1. The experimental section is extremely limited. The manuscript only compares the proposed GCOS method with VOS and a no-regularization baseline on a few datasets for the main results. If the authors aim to position GCOS as an improvement over VOS, they should conduct evaluations on a broader range of benchmarks and tasks, similar to VOS including object detection. Moreover, VOS includes comparisons with multiple baselines, whereas the current experiments are insufficient to substantiate the claimed effectiveness of GCOS.

2. The authors note that the online calibration stage inherently violates the exchangeability assumption central to conformal prediction. The subsequent post-training calibration mitigates this theoretical limitation. As a result, the conformal component of GCOS lacks full statistical validity, weakening the theoretical rigor of the proposed framework.

**Questions:**

1. What is the computational overhead introduced by GCOS, particularly compared to VOS, given its additional PCA computations and conformal prediction?

2. How well does GCOS generalize to diverse OOD scenarios, like object detection or other tasks?

---

> ### Author Response · Authors · 2025-11-26
> **Important Changes Made - Additional Experiments Added**
>
> We want to thank the reviewer for raising several valid concerns and noting several areas for improvement of our paper. Based on the reviews, we made several important additions and obtained new results from extra experiments and ablations.
> We apologize for the delayed response - we took time to run more experiments and verify the correctness/validity of the obtained results. For the reviewer’s convenience, we highlight the modified/added text in the revised PDF file. Specifically, the main modifications made in the revised version:
>
> - Included more baselines in the main results section (Table 1): several classical approaches (MSP, MaxLogit), more modern regularization-based methods (ReAct, ViM), and one SOTA (NPOS). Our approach is superior to all the baselines.
>
> - Renamed Section 6 to "Moving Towards OOD Detection with Statistical Guarantees" to better highlight an important research direction. While Contribution 3 (conformal classification head with formal statistical guarantees) shows preliminary results and represents extended future work, we believe this direction warrants community attention. The main validated approach, Contributions 1 and 2 (geometric outlier synthesis with conformal shell heuristic and contrastive regularization), achieves the superior performance reported in Table 1.
>
> - Added multi-seed validation to assess score variability (Table 8 in Appendix E2).
>
> - Added object detection evaluation (in contrast to multi-class classification in the main part of the paper) in Table 9 in Appendix E3. With no hyperparameter tuning, our approach remains on par with SOTA and even dominates on the OpenImages dataset.
>
> - Added a small ablation study on the necessity of the conformal shell during synthesis in Table 11 in Appendix F2. When $\alpha$ is sampled outside the conformal shell bounds, performance degrades.
>
> - Extended the literature review section.
>
> - Fixed typos.
>
>
> Now, let us address the questions and comments raised by the reviewer:
>
> - "The experimental section is extremely limited". We include modern approaches (NPOS, ReAct, ViM) and show that Near-OOD datasets are quite challenging and GCOS is superior. Note: We also considered several versions of ODIN (e.g., Liang et al., 2018), which resulted in inferior scores in Table 1 across all datasets and were thus excluded, and another SOTA method such as NCIS (Doorenbos et al., 2024), which requires diffusion-based generation and is significantly more computationally demanding than VOS or GCOS and thus could not finish training within the rebuttal period.
>
> - "conduct evaluations on a broader range of benchmarks". While the classification task is the most common in this kind of literature, the VOS paper indeed conducted evaluations on the object detection task. We follow this reviewer’s recommendation and provide evaluation on object detection in Appendix E3 and note competitive performance.
>
> - "What is the computational overhead introduced by GCOS". Since synthesis is applied in the feature space preceding the final classification head, where data is assumed to be linearly separable and has fewer dimensions than the original image, PCA introduces marginal overhead compared to VOS.
>
> - "GCOS lacks full statistical validity". The main results in Table 1 are based on Contributions 1 and 2 (geometric outlier synthesis with conformal shell heuristic), which do not claim formal statistical guarantees. We use the conformal shell as a practical heuristic for adaptive synthesis control, and it demonstrably improves OOD detection over all baselines. Contribution 3 (conformal classification head with formal statistical guarantees) is separate work presented as extended future work with preliminary results. We acknowledge that while our heuristic approach is validated and effective, establishing formal statistical validity remains an important direction for future research.
>
>
> We again want to thank the reviewer for pointing out the valid weaknesses, which allowed us to address them and improve this manuscript. Based on additional results and consistent improvement in scores over baselines, we respectfully ask the reviewer to reconsider the final rating, and we stay open to further discussion and clarifications.
>
>
> Changes: L318-L377, L401, L975, L1016-L1056, L1061, L1099-L1108

---

### Official Review · Reviewer_KjEs · 2025-10-30

**Soundness:** 2
**Presentation:** 2
**Contribution:** 2
**Rating:** 2
**Confidence:** 4

**Summary:**

The authors propose to combine conformal learning with outlier synthesis. The outliers are synthesized by sampling from off-manifold directions, as defined by low-variance principal components. During training, these outliers are encouraged by a regularization loss to move farther away from the class centers. At inference time, the standard energy-based metric is used to detect outliers. The method is compared against VOS on some near-OOD experiments.

**Strengths:**

- The application of conformal prediction to outlier synthesis is interesting.
- Sampling in off-manifold directions is a sensible approach.
- The method is clearly explained, and the idea of using conformal prediction during inference is interesting.

**Weaknesses:**

The method fails to mention a number of papers published since VOS that effectively address its Gaussian assumption shortcoming. One example, NPOS [1], is even explicitly cited as justification for the weaknesses of VOS, yet it is not compared against, despite addressing the exact same pitfalls. Other examples include Dream-OOD [2], BOOD [3], and NCIS [4], which all (especially [3,4]) take the same approach of off-manifold sampling. The paper should both compare to and discuss these methods.

Furthermore, while moving towards Near-OOD experiments for evaluation is sensible, the method should still be compared on standard experiments (such as far-OOD on ImageNet) to be able to place it within the wider field. The chosen Near-OOD experiments are also very simple, such as MVTec, which has been almost perfectly solved for a long time [5], and the models reach close to perfect scores on both MNIST and Stanford Dogs. More standard Near-OOD experiments would be appropriate, such as ImageNet:SSB-Hard and CIFAR100:CIFAR10.

Overall, due to these aspects, the current manuscript does not convincingly demonstrate that the proposed method is superior to the other methods proposed for the same purpose.

[1] Tao, Leitian, et al. "Non-parametric outlier synthesis." International Conference on Learning Representations 2023.
[2] Du, Xuefeng, et al. "Dream the impossible: Outlier imagination with diffusion models." Advances in Neural Information Processing Systems 36 (2023): 60878-60901.
[3] Liao, Qilin, et al. "BOOD: Boundary-based Out-Of-Distribution Data Generation." International Conference on Machine
Learning 2025.
[4] Doorenbos, Lars, Raphael Sznitman, and Pablo Márquez-Neila. "Non-Linear Outlier Synthesis for Out-of-Distribution Detection." arXiv preprint arXiv:2411.13619 (2024).
[5] Roth, Karsten, et al. "Towards total recall in industrial anomaly detection." Proceedings of the IEEE/CVF conference on computer vision and pattern recognition. 2022.

**Questions:**

- The authors mention multiple times the need to tune energy-based scores on validation data. However, their approach utilizes two separate calibration sets. Why is this not a similar weakness for the proposed method?
- What are the mean and standard deviation of the results over multiple runs?
- The sentence "we start with the discussion of Virtual Outlier Synthesis (VOS) (Du et al., 2022), thereby shaping more robust decision boundaries," on L38-40 is grammatically incorrect. The same holds for L233-235.

---

> ### Author Response · Authors · 2025-11-26
> **Important Changes Made - Additional Experiments Added**
>
> We want to thank the reviewer for raising several valid concerns and noting several areas for improvement of our paper. Based on the reviews, we made several important additions and obtained new results from extra experiments and ablations.
> We apologize for the delayed response - we took time to run more experiments and verify the correctness/validity of the obtained results. For the reviewer’s convenience, we highlight the modified/added text in the revised PDF file. Specifically, the main modifications made in the revised version:
>
> - Included more baselines in the main results section (Table 1): several classical approaches (MSP, MaxLogit), more modern regularization-based methods (ReAct, ViM), and one SOTA (NPOS). Our approach is superior to all the baselines.
>
> - Renamed Section 6 to "Moving Towards OOD Detection with Statistical Guarantees" to better highlight an important research direction. While Contribution 3 (conformal classification head with formal statistical guarantees) shows preliminary results and represents extended future work, we believe this direction warrants community attention. The main validated approach, Contributions 1 and 2 (geometric outlier synthesis with conformal shell heuristic and contrastive regularization), achieves the superior performance reported in Table 1.
>
> - Added multi-seed validation to assess score variability (Table 8 in Appendix E2).
>
> - Added object detection evaluation (in contrast to multi-class classification in the main part of the paper) in Table 9 in Appendix E3. With no hyperparameter tuning, our approach remains on par with SOTA and even dominates on the OpenImages dataset.
>
> - Added a small ablation study on the necessity of the conformal shell during synthesis in Table 11 in Appendix F2. When $\alpha$ is sampled outside the conformal shell bounds, performance degrades.
>
> - Extended the literature review section.
>
> - Fixed typos.
>
>
> Now, let us address the questions and comments:
>
> - "fails to mention a number of papers". The literature review and baseline methods sections were extended.
>
> - "The paper should both compare to and discuss these methods". We include modern approaches (NPOS, ReAct, ViM) and show that Near-OOD datasets are quite challenging and GCOS is superior. Note: We also considered several versions of ODIN (e.g., Liang et al., 2018), which resulted in inferior scores in Table 1 across all datasets and were thus excluded, and another SOTA method such as NCIS (Doorenbos et al., 2024), which requires diffusion-based generation and is significantly more computationally demanding than VOS or GCOS and thus could not finish training within the rebuttal period.
>
> - "Near-OOD experiments are also very simple". In extended Table 1, we show that many methods struggle on these datasets. The SOTA approach, NPOS, with default hyperparameters completely fails on C-MNIST and Retinopathy, which presented challenges for these methods. Also, while Colored MNIST was intended as a toy dataset, we notice that many methods tend to "overfit" to one modality of information (either the color or the shape of the digit) and thus struggle on OOD examples.
>
> - "MVTec, which has been solved almost perfectly". We included MVTec for completeness as a toy dataset; we acknowledge it is less challenging than our other benchmarks.
>
> - "More standard Near-OOD experiments would be appropriate (ImageNet:SSB-Hard and CIFAR100:CIFAR10)". We acknowledge the importance of standard benchmarks for comparability. However, for ImageNet:SSB-Hard and CIFAR100:CIFAR10, there is significant class overlap between ID and OOD sets, which creates ambiguity in evaluation: samples from overlapping classes cannot be definitively labeled as OOD. The non-overlapping classes in these splits tend to exhibit Far-OOD characteristics rather than Near-OOD, which is the focus of our work. We note that we have included object detection evaluation (Appendix E3) on standard benchmarks (Pascal VOC with COCO/OpenImages OOD), which demonstrates GCOS's effectiveness on established evaluation protocols.
>
> - "two separate calibration sets". Contribution 3 (conformal classification head) requires a second calibration set to maintain statistical guarantees, as discussed in Appendix C. However, the main results in Table 1 rely only on Contributions 1 and 2, which use a single calibration set.
>
> - "mean and standard deviation of the results". As requested, we conducted additional multi-seed validation in Appendix E2. Besides, we also evaluated GCOS on the object detection task in Appendix E3 and report robust performance.
>
> We again want to thank the reviewer for pointing out the valid weaknesses, which allowed us to address them and improve this manuscript. Based on additional results and consistent improvement in scores over baselines, we respectfully ask the reviewer to reconsider the final rating, and we stay open to further discussion and clarifications.
>
> Changes: L318-L377, L401, L975, L1016-L1056, L1061, L1099-L1108

---

### Official Review · Reviewer_CQMq · 2025-11-01

**Soundness:** 3
**Presentation:** 3
**Contribution:** 2
**Rating:** 4
**Confidence:** 4

**Summary:**

This paper introduces Geometric Conformal Outlier Synthesis (GCOS), a training-time framework aimed to improve out-of-distribution (OOD) detection and overconfidence. The core idea is to synthesize virtual outliers in the feature space that
respect the learned geometry of the in-distribution (ID) data. This should improve over simple parametric (e.g., Gaussian) assumptions of prior work like VOS.

The GCOS synthesis process has two stages: Direction by applying PCA to the training features to identify geometrically-informed, off-manifold directions, and Magnitude during which a "Conformally-Inspired Shell" is used to adaptively control the synthesis magnitude. The shell is defined by the empirical quantiles (e.g., 95th and 99th) of a nonconformity score (Mahalanobis distance) from a calibration set.

The goal is to generate hard outliers that are not trivially easy from ID data. This synthesis is paired with a contrastive regularization loss. The authors show that GCOS improves OOD detection on several Near-OOD benchmarks compared to a baseline and
VOS.

**Strengths:**

- one clear strength is the move away from simple parametric assumptions for outlier synthesis. Using PCA to find low-variance directions is an intuitive and non-parametric way to explore the odd regions of the learned feature space.

- the paper argues  that the choice of outlier synthesis is key, and that outliers should not be too easy or too hard. The use of a shell based on quantiles of the Mahalanobis score for adaptively calibrating this difficulty based on the model’s current state is original, well justified and, in my opinion, clever

- a positive point is also the focus on near-OOD datasets which are harder, more critical for real world tasks, and the good performance is encouraging

**Weaknesses:**

- the contribution feels incremental, by combining several existing concepts : virtual outlier synthesis, using PCA for geometric analysis, and using Mahalanobis distance for OOD detection. GCOS is a novel combination of these parts, but it does not feel like a fundamentally new approach. In its defense, the combination is well justified and seems effective.- the main results (Table 1) only compare GCOS against VOS and a No Reg. baseline. Authors should ideally benchmark their method against related modern SOTA OOD detection methods on Near-OOD datasets, completing Table 1. While the authors emphasis on Near-OOD is well-motivated, a discussion to support the fact that a method generalizing well to Near-OOD will also perform well on Far-OOD would be appreciated.

 - the central claim is that difficult outliers are superior for training. To prove that this claim is grounded, the authors should compare the strategy to a trivial synthesis method (e.g., generating points with a large range of fixed $\alpha$, or sampling random vectors near the origin).

- the method rests on two strong assumptions, linearity and cluster separation. Using PCA assumes the feature manifold is locally linear and that linear paths along low-variance eigenvectors are meaningful directions. This is a strong structure assumption for deep, high-dimensional feature spaces. Given the known nonlinear nature of deep features, a discussion on the validity of this local linearity assumption or its potential limitations for GCOS is necessary. Secondly, the use of Conformal in the title is questionable. The synthesis method is inspired by conformal prediction but provides no statistical guarantees. The actual formal conformal hypothesis testing (Contribution 3) is presented as Future Work and yields mixed results and can collapse to a nearly random classifier.

**Questions:**

- Could the authors provide results comparing GCOS against one or two more related modern SOTA methods (selected by themselves) on the Near-OOD benchmarks from Table 1 ?

- Can the authors provide an ablation study comparing GCOS to a trivial synthesis baseline (e.g., synthesis with a very large range of $\alpha$, or sampling random latent vectors far from the class custers) to demonstrate that the hard negative calibration from the conformal shell is truly necessary?

- How does GCOS handle cases where class manifolds are adjacent ? What prevents the hidden representation sampled from Class A generating a sample that is easily mistaken for Class B, thereby potentially confusing the classifier? Could they also discuss datasets with high uncertainty possibly leading to hard linear separation in the feature space ?

**Details Of Ethics Concerns:**

nothing concerning

---

> ### Author Response · Authors · 2025-11-26
> **Important Changes Made - Additional Experiments Added**
>
> We want to thank the reviewer for giving a good rating and noting several areas for improvement of our paper. Based on the reviews, we made several important additions and obtained new results from extra experiments and ablations.
> We apologize for the delayed response - we took time to run more experiments and verify the correctness/validity of the obtained results. For the reviewer’s convenience, we highlight the modified/added text in the revised PDF file. Specifically, the main modifications made in the revised version:
>
> - Included more baselines in the main results section (Table 1): several classical approaches (MSP, MaxLogit), more modern regularization-based methods (ReAct, ViM), and one SOTA (NPOS). Our approach is superior to all the baselines.
>
>
> - Renamed Section 6 to "Moving Towards OOD Detection with Statistical Guarantees" to better highlight an important research direction. While Contribution 3 (conformal classification head with formal statistical guarantees) shows preliminary results and represents extended future work, we believe this direction warrants community attention. The main validated approach, Contributions 1 and 2 (geometric outlier synthesis with conformal shell heuristic and contrastive regularization), achieves the superior performance reported in Table 1.
>
> - Added multi-seed validation to assess score variability (Table 8 in Appendix E2).
>
> - Added object detection evaluation (in contrast to multi-class classification in the main part of the paper) in Table 9 in Appendix E3. With no hyperparameter tuning, our approach remains on par with SOTA and even dominates on the OpenImages dataset.
>
> - Added a small ablation study on the necessity of the conformal shell during synthesis in Table 11 in Appendix F2. When $\alpha$ is sampled outside the conformal shell bounds, performance degrades.
>
> - Extended the literature review section.
>
> Now, let us address the questions and comments:
>
> - "The method rests on linearity and cluster separation". Our method operates in the penultimate feature space, immediately before the final classification layer. At this layer, the data should be linearly separable for the classification head to succeed on the main task. We acknowledge that in cases where the pretrained model itself struggles with classification (e.g., poorly trained networks or highly entangled representations), the local linearity assumption may be weaker. However, our empirical results across multiple datasets demonstrate that this assumption holds sufficiently in practice for well-trained models.
>
> - "Contribution 3 yields mixed results". This is true. As mentioned above, we aim to highlight this important research direction; we show that in some scenarios it is feasible and hope that more research will try to solve the problem we initiate here.
>
> - "More related modern SOTA methods". We include modern approaches (NPOS, ReAct, ViM) and show that Near-OOD datasets are quite challenging and GCOS is superior. Note: We also considered several versions of ODIN (e.g., Liang et al., 2018), which resulted in inferior scores in Table 1 across all datasets and were thus excluded, and another SOTA method: NCIS (Doorenbos et al., 2024), which requires diffusion-based generation and is significantly more computationally demanding than VOS or GCOS and thus could not finish training within the rebuttal period. We believe our comparison with NPOS, ReAct, and ViM provides a comprehensive evaluation against modern methods.
>
> - "Provide trivial synthesis baseline". We conducted a small experiment where, while outliers are still generated along a small principal component, $\alpha$ was within a random fixed range, and we report a drop in performance in Appendix E2.
>
> - "... cases where class manifolds are adjacent". GCOS prevents confusion between adjacent class manifolds primarily by sampling along low-variance principal components, which are typically orthogonal to the high-variance discriminative directions that separate classes. This ensures synthetic points move "off-manifold" rather than traversing the inter-class decision boundary. Furthermore, our contrastive regularization loss is class-agnostic in its penalty: it maximizes the nonconformity of a synthetic sample with respect to any class ($\min_{k}$ in Eq. 5). If a sample generated from Class A inadvertently falls near Class B’s manifold, the loss penalizes its low nonconformity score against Class B, forcing the decision boundary to tighten around both clusters.
>
> We again want to thank the reviewer for pointing out the weaknesses, which allowed us to improve this manuscript. Based on additional results and consistent improvement in scores over baselines, we respectfully ask the reviewer to reconsider the final rating, and we stay open to further discussion and clarifications.
>
>
>
> Changes: L318-L377, L401, L975, L1016-L1056, L1061, L1099-L1108

---

### Meta-Review · Area_Chair_tXk2 · 2025-12-30

**Summary:**

Reviewers agreed the paper’s core idea—synthesizing latent-space outliers along low-variance PCA directions and controlling “hardness” via a conformal-quantile shell— is sensible and generally well explained (CQMq, KjEs), and could improve near-OOD robustness beyond VOS. The main concerns driving skepticism were (i) perceived incremental novelty (CQMq, WDSt), (ii) initially limited and nonstandard experimental coverage and missing comparisons to post-VOS methods (KjEs, WDSt), and (iii) the “conformal” framing: the shell is heuristic and does not itself provide statistical validity, while the more formal conformal inference head is preliminary/mixed (CQMq, WDSt).

**Reviewer Concerns:**

The rebuttal addressed some of the experimental critiques: the authors expanded Table 1 with additional baselines including a strong modern reference point (NPOS) plus ReAct/ViM (CQMq, KjEs, WDSt), added multi-seed variability (addressing KjEs), provided an object detection evaluation (addressing WDSt’s breadth concern and KjEs’s “VOS-style” task coverage), and added ablations indicating the conformal-shell bounds matter (addressing CQMq’s request for “trivial synthesis” controls).

The main outstanding issues are that some key contemporary comparators cited by KjEs (e.g., diffusion/nonlinear synthesis lines such as Dream-OOD/BOOD/NCIS) are still not directly evaluated, the benchmark suite remains somewhat debated (standard far-OOD / canonical near-OOD splits vs. authors’ choices), and the “conformal” aspect remains primarily inspirational for the synthesis component (with formal guarantees only in the exploratory head, where results are acknowledged as mixed).

**Reviewer Scores:**

CQMq (4) would plausibly move to a weak accept given the added baselines, the shell ablation, and the broader validation (multi-seed + detection), while still viewing the conceptual novelty as a well-motivated combination rather than a fundamentally new paradigm. KjEs (2) is likely to increase modestly (or stay the same) because the rebuttal directly answers the biggest critique—missing post-VOS baselines—by adding NPOS/ReAct/ViM plus reporting variance and an object-detection setting, though they may remain unconvinced absent head-to-head results vs Dream-OOD/BOOD/NCIS and more standard benchmark placement. WDSt (2) would likely stay the same, as they are likely to still discount the “conformal” framing and view the contribution as incremental.

---

### Decision · Program_Chairs · 2026-01-26

Reject